# Histopathological and Virological Findings of a Penile Papilloma in a Japanese Stallion with Equus Caballus Papillomavirus 2 (EcPV2)

**DOI:** 10.3390/pathogens13070597

**Published:** 2024-07-19

**Authors:** Eri Uchida-Fujii, Yusei Kato, Takanori Ueno, Yasuko Numasawa, Shigeki Yusa, Takeshi Haga

**Affiliations:** 1Division of Infection Control and Disease Prevention, Graduate School of Agricultural and Life Sciences, The University of Tokyo, 1-1-1 Yayoi, Bunkyo-ku, Tokyo 113-8657, Japan; eri-uf@g.ecc.u-tokyo.ac.jp (E.U.-F.);; 2Microbiology Division, Equine Research Institute, Japan Racing Association, 1400-4 Shiba, Shimotsuke 329-0412, Tochigi, Japan; 3The Japan Bloodhorse Breeders’ Association Shizunai Stallion Station, 517 Shizunaitahara, Shinhidaka-cho 056-0144, Hokkaido, Japan; 4Laboratory of OSG Veterinary Science for Global Disease Management, Graduate School of Agricultural and Life Sciences, The University of Tokyo, 1-1-1 Yayoi, Bunkyo-ku, Tokyo 113-8657, Japan

**Keywords:** *Equus caballus* papillomavirus 2, genital tumor, Thoroughbred, papillomas, penile tumor

## Abstract

Equus caballus papillomavirus 2 (EcPV2) is known to cause genital neoplasms in horses. However, reports on EcPV2 in Japan and Asia are limited. Herein, we present the histopathological and virological findings of the first reported case of an EcPV2-associated penile mass in Japan. The patient was a 22-year-old stallion with a history of breeding in Japan and abroad. Histopathological examination contained RNA in situ hybridization targeting the E6/E7 region and an immunohistochemical approach, and whole-genome sequencing was conducted within the viral examination. Proliferating epidermal cells were observed, and EcPV2 E6/E7 mRNA was detected within the epidermis, which was interpreted as viral papilloma. The detected EcPV2 virus was genetically close to foreign strains and different from the strain previously reported from a Japanese mare. This suggests that various types of EcPV2 might already exist among horses in Japan. Although the mass reported herein was not malignant based on histopathological findings and the absence of recurrence, its presence on the penis would be an obstacle to breeding. These results provide a better understanding of the pathogenesis and diversity of EcPV2.

## 1. Introduction

Equus caballus papillomaviruses are currently classified into 10 types, Equus caballus papillomavirus (EcPV) 1 to EcPV10 [1]. Among them, *Equus caballus* papillomavirus 2 (EcPV2) is known to cause genital squamous cell carcinoma (SCC) in mares and stallions [2,3]. Although the relationship between papilloma infection and cutaneous neoplasms is common among companion animals, EcPV2 is distinctive in that it produces neoplasms on mucosal surfaces [4]. The EcPV2 genome contains open reading frames for oncoproteins E6 and E7, regulatory proteins E1, E2, and E4, and capsid proteins L1 and L2 [5]. Among the early genes in EcPVs, E6 is considered an oncogene of major interest [6]. EcPV2 was reported to be detected in 57% of genital papillomas and 29–100% of genital SCCs in horses, even though 2–24% of the normal genital tissues were EcPV2-positive [5,6,7,8,9]. EcPV2 has also been detected in the papillomas of cutaneous [10] and gastric tumors [11]. It has been suggested that EcPV2 could persistently infect the genital tissues of horses and be transmitted by sexual contact. Other transmission routes, including insect-transmitted infection, are also assumed to be present [7,12]. Breeding might be considered one of the risk factors for EcPV2 transmission [13].

EcPV2-associated genital tumors have been reported in most regions of the world [5]. In Asia, there have been few reports on EcPV2-associated genital tumors [14,15] and limited reports on EcPV2 positivity in asymptomatic horses [16]. In Japan, only two EcPV2 cases have been reported to date, one with a laryngeal tumor [17] and the other in a mare with a genital tumor [15]. Due to the limited reports on EcPV2 in Asia, the pathological features, effects of infection on horses, epidemiological features, infection status, and genetic diversity of EcPV2 remain unclear.

We encountered a stallion with a penile mass associated with EcPV2, the first reported case of a male with EcPV2 in Japan. Here, we present the histopathological and virological findings in this case of EcPV2 infection.

## 2. Detailed Case Description

### 2.1. Case

The patient was a 22-year-old retired Thoroughbred stallion. He was used as a racehorse in Ireland until he reached three years old, then served as a stallion in the United States and Australia for 5 years, followed by 2 years in the United States and Argentina. He was then imported to Japan, where he served as a stallion for 10 years. The patient mated with dozens to three hundred mares per year until 3 years before the resection of the mass. After retiring as a stallion, he remained in Japan. Depigmented plaques had been observed in his penis for around 12 years before the resection of the mass, and finally a 4 cm × 4 cm mass on the top of the penis was observed without pain or deterioration of the general condition (Figure 1). The mass was resected under sedation and local anesthesia. The half-piece of the mass was fixed with 10% neutral buffered formalin for histopathological testing, and the resting half was kept at −20 °C for virological testing. Twelve months after mass extraction, no recurrence was observed, and the patient was in good general condition.

### 2.2. Histopathological Findings

The formalin-fixed mass was embedded in paraffin wax and stained with hematoxylin and eosin (HE) for light microscopy. To confirm the localization of EcPV2 in the mass tissues, RNA in situ hybridization (ISH) targeting the E6/E7 region was performed. The ISH probe was designed with reference to the EcPV2 genome sequence obtained below and designed by a commercial company (Advanced Cell Diagnostics Inc. [ACD], Newark, CA, USA). The expression of Ki-67 and p53, associated with cell proliferation and tumor suppression, respectively, was examined using immunohistochemistry. The primary antibodies were anti-Ki-67 rabbit polyclonal antibody (Abcam, Cambridge, UK; dilution 1:500) and anti-p53 mouse monoclonal antibody (Santa Cruz Biotechnology, Santa Cruz, CA, USA; dilution 1:100). Deparaffinized tissue sections were subjected to antigen retrieval in a pressure steam chamber (DAKO Pascal, DAKO, Glostrup, Denmark) with Target Retrieval Solution pH9 (DAKO, for anti-Ki-67 antibody) or citrate buffer (Abcam, for anti-p53 antibody) for the prescribed time. The sections were treated with 0.03% H_2_O_2_ in methanol to quench endogenous peroxidase activity, followed by incubation with a protein-blocking solution (Abcam) to block nonspecific binding sites. Sections were then incubated with primary antibodies for 1 h at room temperature. The immunoperoxidase polymer method (Histofine Simple Stain MAX-PO (MULTI), Nichirei Bioscience, Tokyo, Japan) was used.

In the HE-stained section, proliferating epidermal cells arranged in a papillary fashion were observed, supported by fibrovascular stroma in the head of the mass (Figure 2A). Extensive parakeratosis was observed, and swollen epidermal cells were scattered superficially. In some areas, owing to hyperplasia of the basal cells, the basal cell arrangement was multilayered, or the layered structure was obscured. Small, round dyskeratotic foci were observed in the middle and deep epidermal layers. Mild lymphocyte infiltration was observed in the subepidermal stroma. In the stalk part of the mass, the epidermis was prominently thickened with orderly maturation and formed broad-rate pegs. Superficial epidermal cells with swelling, cytoplasmic vacuolation, and pyknosis were occasionally observed, and mild parakeratosis was observed in some areas (Figure 2B). The basal cell arrangement was standard, and moderate infiltration of lymphocytes and proliferation of fibroblasts were observed in the subepidermal stroma.

In the ISH assay, EcPV2 E6/E7 mRNA was detected at any location within the epidermis. Basal and spinous cells in the middle or deep epidermis showed fine-granular signals scattered throughout the nucleus and cytoplasm. Particularly in the stalk, few spinous cells with strong diffuse nuclear signals were observed (Figure 2C,D).

Immunohistochemically, several basal cells were positive for Ki-67 in both parts of the mass. In addition, some deep epidermal cells other than basal cells were also positive in the head. Epidermal cells in the head expressed p53 in a distribution similar to that of Ki-67, whereas fewer basal cells expressed p53 in the stalk (Figure 2G,H).

The mass was diagnosed as viral papilloma based on the obtained histopathological findings.

### 2.3. Virological Findings

Virological tests were performed as previously described [15]. DNA was obtained using QIAamp DNA Mini Kit (QIAGEN, Hilden, Germany) according to the manufacturer’s instructions. Conventional PCR was performed to detect EcPV1, EcPV2, bovine papillomavirus 1 (BPV1), and BPV2, using the DNA polymerase KOD FX Neo (Toyobo, Otsu-shi, Shiga, Japan). The used primers, 498 EcPV1 L1 for EcPV1, 445 EcPV2 L1 for EcPV2, and BPV1/2 L1 subA modified for BPV1 and BPV2, and incubation protocol are shown in Appendix A. A positive band for EcPV2 was observed, whereas bands for EcPV1 and BPV1 were negative.

The whole genome of EcPV2 was sequenced as previously described [15]. Briefly, PCR amplicons using primers EcPV2 415 (F)/EcPV2 3115 (R), EcPV2 2714 (F)/EcPV2 5705 (R) and EcPV2 5264 (F)/EcPV2 543 (R) (Appendix A) in 1.2% agarose gel were gel-purified with the NucleoSpin Gel and PCR Clean-Up kit (Macherey-Nagel, Düren, Germany). Direct sequencing was obtained by Sanger Sequencing Service (AZENTA, Tokyo, Japan). The obtained sequence was shown as LC810153 in the DDBJ database. The nucleotide sequence was compared to the reported EcPV2 strains using the NCBI BLAST tool (https://blast.ncbi.nlm.nih.gov/Blast.cgi, accessed on 31 December 2023). Phylogenetic analysis of the L1 and E6 regions with EcPV2 strains obtained from the NCBI GenBank https://www.ncbi.nlm.nih.gov/genbank/ (accessed on 31 December 2023) was performed using MEGA 11 [18].

The nucleotide sequence of the L1 region of our isolate was 98.67–100% matched compared to the reported 14 EcPV2 strains with 1500 bp nucleotide sequences (Table 1). Our isolate displayed a 100% identity with strains reported from Italy. Additionally, a single amino acid mutation (p. Ala280Thr) was observed in the strains, except in 9 out of the 14. In the phylogenetic tree based on L1, our isolate is included in the cluster with EcPV2 isolates from Italy, which are separated from the cluster with the former Japanese EcPV2 isolate from a mare’s genital tumor (accession no. LC612601) (Appendix A). In the phylogenetic tree based on E6, our isolate is included in a cluster with EcPV2 isolates from Australia, Italy, Belgium, and China, and is also separated from the cluster with the former Japanese EcPV2 isolate from the genital tumors of mares (Appendix A).

## 3. Discussion

To our knowledge, this is the first report of an EcPV2-associated genital mass in a Japanese stallion. The histopathological findings contribute to the understanding of the pathological status of EcPV2-associated penile neoplasms. Furthermore, information on the whole genome of our strain will contribute to understanding the genomic variations in EcPV2.

Ramsauer et al. showed that an EcPV2-associated penile mass could be histopathologically distinguished from an early stage, including hyperplasia and papilloma, to the late stage, including in situ carcinoma and SCC [19]. In our case, pathological findings of papilloma with proliferating epidermal cells, granular signals detected via ISH, and p53-positive deep epithermal cells were observed especially in the head of the mass compared to the stalk, which was consistent with the previously reported findings of papilloma [19]. The nuclear staining images in ISH for E6/E7 were also observed in human papillomavirus-associated tumors, which is thought to be associated with the life cycle of human papillomavirus; DNA episome synthesis increases and single-stranded human papillomavirus (HPV) occurs in the productive phase of HPV [20].

Genetic variation in E6 of EcPV2 has been suggested to be unrelated to the site or malignancy of the tumor [5]. The sequence obtained from E6 of our isolate showed an amino acid deletion (p. Arg72del), which was not typical for our isolate, and its impact on the function of E6 was unknown. In this study, the phylogenetic tree obtained based on E6 and L1 could not elucidate the geographic origins of strains, due to the limitation that the genetic information of EcPV2 has been reported from a limited number of countries and may not accurately reflect the global genetic variation in EcPV2. In contrast, our previous study suggested that bovine papillomavirus type 1 (BPV1), which is known to be associated not only with papilloma in bovines but also with sarcoid formation in equine skin, has a genetic variation related to its geographic origin [21]. This difference could be due to the differences in the transmission routes of BPV1 and EcPV2. BPV1 is thought to be transmitted directly among nearby horses and bovines, while sexual transmission is thought to be one of the transmission routes of EcPV2 [7,21]. Horses, especially Thoroughbred racehorses, often breed across countries. The phylogenic analysis showed that our isolate might be genetically close to isolates from Australia, Europe, and China. The case might have experienced the EcPV2 infection prior to onset, and EcPV2 could have been reactivated due to immune suppression or aging. The history of overseas breeding may be associated with EcPV2 infection. Breeding in Japan may also have led to EcPV2 infection; mares from abroad are sometimes used for breeding in Japan. The breeding with asymptomatic carriers in Japanese mares may be one of the transmission routes of our isolate. It was noted that our isolate was genetically different from the EcPV2 strain reported from a Japanese mare. Several genetic types of EcPV2 may already exist among horses in Japan, despite the limited information on EcPV2 in the country. The infection status and genetic variation in EcPV2 should be surveyed among mares and stallions in Japan to understand the EcPV2 status in Japanese horses.

In our case, the mass showed no malignancy due to the patient’s good general condition and its absence of reoccurrence. Histopathological findings also suggested that our patient’s tumor remained in an early stage compared with the SCC in a previous report [19]. Greenwood et al. suggested that EcPV2 status in SCCs is not associated with horse survival rates [6]. Even though EcPV2 infection is not often prognostic for horse life, the mass on the penis itself would be an obstacle to breeding, which would compromise the productivity of stallions. Furthermore, EcPV2 in the genital region could be transmitted through breeding, posing a risk of EcPV2 infection for other horses. The association of EcPV2 with neoplasms in the mucosa, especially in the genitalia, is noteworthy for EcPV2, and the impact on the genital mass with EcPV2 in horse breeding should be carefully evaluated.

## 4. Conclusions

A case of an EcPV2-associated penile mass was reported in Japan. Genetic features of the detected EcPV2 virus suggested a possible association with foreign strains and suggested that various types of EcPV2 might already exist among horses in Japan. These results provide a better understanding of the pathogenesis and diversity of EcPV2.

## Figures and Tables

**Figure 1 pathogens-13-00597-f001:**
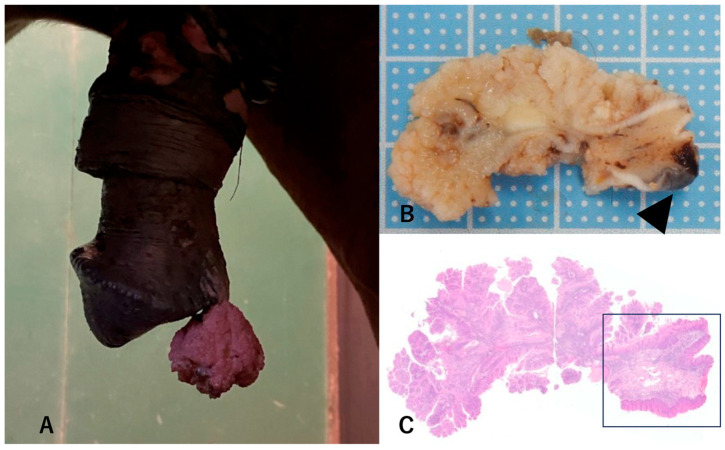
Penile mass in a 22-year-old Thoroughbred stallion. (**A**) Mass before extraction. (**B**) Slit surface of the formalin-fixed mass. The resection area is indicated by arrows. (**C**) Loupe view of the mass. Epidermal hyperplasia was observed in the stalk (indicated by a square) and papillomas in the head. Hematoxylin and eosin (HE) staining.

**Figure 2 pathogens-13-00597-f002:**
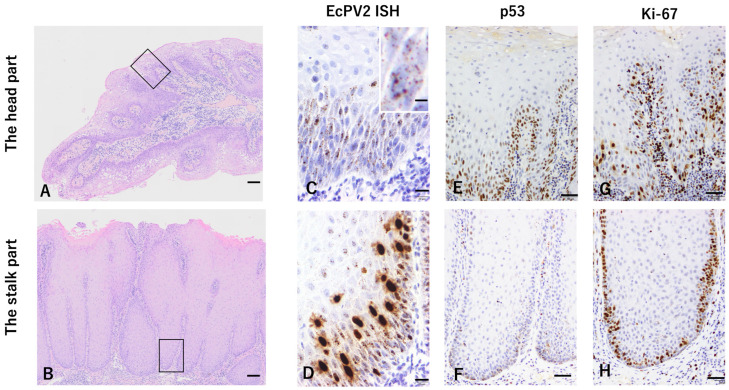
Photomicrographs of Equus caballus papillomavirus type 2 (EcPV2)-associated penile mass. Findings pertaining to the head of the mass are shown in the top row, and findings pertaining to the stalk of the mass are in the bottom row. (**A**) Proliferated epidermal cells were observed in a papillary fashion supported by a fibrovascular stroma. The area shown in (**C**,**E**,**G**) is indicated with a square. Hematoxylin and eosin (HE) staining was conducted. Bar, 100 μm. (**B**) The epidermis prominently thickened with orderly maturation and formed broad rate pegs. The area shown in (**D**,**F**,**H**) is indicated with a square. HE staining. Bar 100 μm. (**C**) Basal and spinous cells located in the middle or deep epidermis with the scattered fine-granular signal. In situ hybridization (ISH) was performed. Bar, 20 μm. The inset shows the cell with the scattered fine-granular signal. ISH. Bar, 5 μm. (**D**) Spinous cells had strong diffuse nuclear signals. ISH. Bar, 20 μm. (**E**) Some deep epidermal cells and basal cells were positive. Immunohistochemistry. p53. Bar, 50 μm. (**F**) Fewer basal cells were positive. Immunohistochemistry. p53. Bar, 50 μm. (**G**) Some deep epidermal cells and basal cells were positive. Immunohistochemistry. Ki-67. Bar, 50 μm. (**H**) Basal cells were positive. Immunohistochemistry. Ki-67. Bar, 50 μm.

**Table 1 pathogens-13-00597-t001:** Similarity compared with the reported EcPV2 strains, L1 region, 1500 bp.

Accession No.	Reported From	Identity (%)
ON942232.1	Italy	100
ON942233.1	Italy	100
ON989002.1	Italy	100
LC612601.1	Japan	99.73
ON942231.1	Italy	99.73
EU503122.1	United Kingdom	99.67
MW410986.1	China	99.67
NC_012123.1	United Kingdom	99.67
ON989001.1	Italy	99.67
ON988999.1	Italy	99.60
ON989000.1	Italy	99.60
HM461973.1	Switzerland	99.00
MT063186.1	Italy	98.74
MT063185.1	Italy	98.67

## Data Availability

The original contributions presented in the study are included in the article/Appendix A, and further inquiries can be directed to the corresponding author.

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
