# Peer review of "Histopathological and Virological Findings of a Penile Papilloma in a Japanese Stallion with Equus Caballus Papillomavirus 2 (EcPV2)"

_pathogens, 2024, doi:10.3390/pathogens13070597_

Round 1

Reviewer 1 Report

Comments and Suggestions for Authors

The manuscript describes a EcPV2-induced papilloma on the penis of a horse in Japan. The main significance of the report is that this is the first time a penile lesion has been reported in Japan, although these lesions have been reported throughout the world so this is of limited significance in my opinion. This is especially true as neoplasia associated with EcPV2 has been reported in horses in Japan.

Title – instead of penile mass call it a penile papilloma (or even a wart)

Line 20. A papilloma is by definition caused by virus induced epithelial hyperplasia. Therefore saying the lesion has two findings – hyperplasia and papilloma does not make sense. All papillomas will have areas in which the hyperplastic epithelium is folded forming a papilloma and areas around the edges of the lesion in which the hyperplasia is less and therefore not folded. To suggest this is somehow unusual does not make sense.

Line 21. What is the proliferative phase? If the authors mean when there is more marked folding, again this is in every viral papilloma.

Line 24. Papillomas are not neoplastic and so there is no malignancy. If you want to say there is low malignancy you have to classify this as a neoplasm (and provide evidence to support this classification).

Line 32. Make it clear you are talking about types 1-10.

Sentence line 34-35 ‘and neoplastic….’ I’m cannot understand this. Please modify for clarity.

Line 42. As most affected horses are geldings and EcPV2 has been detected in foals, I have not seen any real suggestion or evidence EcPV2 is spread by breeding. This is mentioned multiple times in the manuscript and should be de-emphasized.

Line 44. Why should breeding be considered a risk factor? The reference seems to be BPVs in cattle so this sentence should be removed.

Line 45. EcPV2 tumors have been reported in almost all regions of the world not just Europe as stated.

Line 59. If you want to build a case this was sexually transmitted state when the horse was last used for breeding. If it was years ago it would be less likely maybe.

Histology. Describe the HE findings and then the extra tests. Please note, IHC does not allow expression to be assessed, just the presence of the protein on the cell. Especially true for p53 as mutation prevents breakdown of the protein rather than affecting expression. I’m also unsure why Ki-67 and p53 IHC was done – what do these show in a papilloma?

Histology. The histology pictures are nice, but small. However from what I can see this looks like a typical viral papilloma. I would say the ‘head’ and the ‘stalk’ look very similar just with hyperplasia and folding – as is typical for a papilloma. I would not talk about the ‘stalk’ and ‘head’ as if they are different lesions – they are the same lesion. Also state your diagnosis – a viral papilloma presumably.

In results state what % similarity between L1 of the present EcPV2 and those of sequences. What protein differences were present in L1. As PVs are classified using L1 I do not know why the information about E6 is included. This includes the phylogenic tree – this should be L1 if you want to classify the virus. Again, I would prefer a table comparing L1 sequences of EcPV2 in the world with % similarity – the tree is hard to interpret as the scale bar is too small to see.

Line 154. As before – there is only 1 lesion rather than 2

Line 158. As before every papilloma has an area of less severe hyperplasia around the edges – it has to be this way, you can’t go straight from normal epidermis to papilloma – there is always this transition. I think this paragraph can be removed.

Line 182. I completely agree the main transmission route of EcPV2 is sexual. As before, this virus has been shown to be more common in geldings and has been detected in horses that have never bred and in foals.

Line 185. This is very speculative. It seems there is only 1 other Japanese EcPV2? However, there are lots from elsewhere – until you have more Japanese EcPV2 genomes making any sort of firm conclusions about the origin of the virus cannot be supported.

Line 197. The previous report was a single case – therefore the statement that mares had poor outcomes doesn’t make sense. Most of the conclusions of this paragraph seem to be based on a single papilloma in a stallion and a single SCC in a mare. This isn’t justified.

Line 206. Every papilloma has variable severity of hyperplasia – these are not two different histological findings as stated.

Comments on the Quality of English Language

generally good, just a couple of small reasonably easy to fix things.

Author Response

Thank you for kindly reviewing and supportive suggestions. The revised manuscript highlighted the modified points according to the reviewers’ comments in yellow and highlighted the grammatically modified points in red font. We revised the manuscript as follows.

Comment 1: Title – instead of penile mass call it a penile papilloma (or even a wart)

Response 1: The title was changed to “Histopathological and virological findings on penile papilloma of Japanese stallion with Equus caballus papillomavirus 2 (EcPV2)”

Comment 2: Line 20. A papilloma is by definition caused by virus induced epithelial hyperplasia. Therefore saying the lesion has two findings – hyperplasia and papilloma does not make sense. All papillomas will have areas in which the hyperplastic epithelium is folded forming a papilloma and areas around the edges of the lesion in which the hyperplasia is less and therefore not folded. To suggest this is somehow unusual does not make sense.

Comment 3: Line 21. What is the proliferative phase? If the authors mean when there is more marked folding, again this is in every viral papilloma.

Response 2 and 3: The sentence was revised: “Proliferating epidermal cells were observed and EcPV2 E6/E7 mRNA was detected within the epidermis, which was interpreted as viral papilloma.” 

Comment 4: Line 24. Papillomas are not neoplastic and so there is no malignancy. If you want to say there is low malignancy you have to classify this as a neoplasm (and provide evidence to support this classification).

Response 4: The sentence was revised: “Although the mass reported herein was not malignant based on histopathological findings and the absence of recurrence” The sentence in Discussion section was also revised: “In our case, the mass showed no malignancy due to the patient’s good general condition and absence of reoccurrence.”

Comment 5: Line 32. Make it clear you are talking about types 1-10.

Response 5: The sentence was revised: “Equus caballus papillomaviruses are currently classified as 10 species, Equus caballus papillomavirus (EcPV) 1 to EcPV10 [1].”

Comment 6: Sentence line 34-35 ‘and neoplastic….’ I’m cannot understand this. Please modify for clarity.

Response 6: We revised the sentence: “Although the relationship between papilloma infection and cutaneous neoplasms is common among companion animals, EcPV2 is distinctive in that it produces neoplasms on mucosal surfaces [4].”

Comment 7: Line 42. As most affected horses are geldings and EcPV2 has been detected in foals, I have not seen any real suggestion or evidence EcPV2 is spread by breeding. This is mentioned multiple times in the manuscript and should be de-emphasized.

Response 7: We revised the sentence: “It has been suggested that EcPV2 could persistently infect the genital tissues of horses and be transmitted by sexual contact. Other transmission routes, including insect-transmitted infection, are also assumed [7,12]”

Comment 8: Line 44. Why should breeding be considered a risk factor? The reference seems to be BPVs in cattle so this sentence should be removed.

Response 8: The reference mentioned the EcPV2 infection and the risk factors in the Horses section as below.

“In horses, the etiological roles of Equus caballus papillomavirus 2 (EcPV2) are known and have been widely studied in equine genital neoplasms, including malignant tumors, such as SCCs [27]. Several other EcPV types, such as EcPV3, EcPV4, EcPV7, and EcPV9, have also been identified from equine genital-derived samples and lesions [28,29,30], but these genotypes do not seem to show metastatic behavior, unlike EcPV2 [79]. In HPVs, it is known that sexual transmission could become one of the risk factors for viral infection. In thoroughbred racehorse breeding, artificial breeding, such as by artificial insemination (AI) and embryo transfer (ET), is restricted by the criteria established by the International Stud Book Committee (ISBC). Therefore, it is important to consider the risk of EcPV transmission during natural mating of horses.”

We revised the sentence to be de-emphasized: “Breeding might be considered one of the risk factors for EcPV2 transmission [13].” 

Comment 9: Line 45. EcPV2 tumors have been reported in almost all regions of the world not just Europe as stated.

Response 9: We revised the sentence: “EcPV2-associated genital tumors have been reported in most regions of the world [5].”

Comment 10: Line 59. If you want to build a case this was sexually transmitted state when the horse was last used for breeding. If it was years ago it would be less likely maybe.

Response 10: The case was used as a stallion until 3 years before the mass resection, and depigmented plaques were observed in his penis about 12 years before the mass resection. We add this information in the Case section.

Comment 11: Histology. Describe the HE findings and then the extra tests. Please note, IHC does not allow expression to be assessed, just the presence of the protein on the cell.Especially true for p53 as mutation prevents breakdown of the protein rather than affecting expression. I’m also unsure why Ki-67 and p53 IHC was done – what do these show in a papilloma?

Response 11: We revised the sentence: “In the HE-stained section, proliferating epidermal cells arranged in a papillary fashion were observed, supported by fibrovascular stroma in the head of the mass (Figure 2A).” The paragraph lines 87-95 shows the HE findings, lines 108-110 shows ISH findings, and lines 111-113 shows IHC findings. We tried to evaluate our sample with Ki-67 and p53 IHC to compare to the findings of EcPV2-associated neoplasms reported in doi:10.1186/s12917-019-2097-0.

Comment 12: Histology. The histology pictures are nice, but small. However from what I can see this looks like a typical viral papilloma. I would say the ‘head’ and the ‘stalk’ look very similar just with hyperplasia and folding – as is typical for a papilloma. I would not talk about the ‘stalk’ and ‘head’ as if they are different lesions – they are the same lesion. Also state your diagnosis – a viral papilloma presumably.

Response 12: Figure 2 was enlarged. The sentence “The histopathological findings differed between the head and stalk of the masses (Figure 1 and 2)” was removed. We also add the sentence: “The mass was diagnosed as viral papilloma based on the obtained histopathological findings.” 

Comment 13: In results state what % similarity between L1 of the present EcPV2 and those of sequences. What protein differences were present in L1. As PVs are classified using L1 I do not know why the information about E6 is included. This includes the phylogenic tree – this should be L1 if you want to classify the virus. Again, I would prefer a table comparing L1 sequences of EcPV2 in the world with % similarity – the tree is hard to interpret as the scale bar is too small to see.

Response 13: Similarity with the EcPV2 sequences based on L1 regions were shown in Table 1. The phylogenetic tree based on L1 and E6 were shown as Supplementary Figures. The sentence for method “The nucleotide sequence was compared to the reported EcPV2 strains using the NCBI BLAST tool (https://blast.ncbi.nlm.nih.gov/Blast.cgi).” and the sentence for result “The nucleotide sequence of the L1 region of our isolate was 98.67%–100% matched compared to the reported 14 EcPV2 strains with 1,500 bp nucleotide sequences (Table 1). Our isolate displayed 100% identity with strains reported from Italy. Additionally, a single amino acid mutation (p. Ala280Thr) was observed compared to 9 out of the 14 strains.” were added.

Comment 14: Line 154. As before – there is only 1 lesion rather than 2 Line 158. As before every papilloma has an area of less severe hyperplasia around the edges – it has to be this way, you can’t go straight from normal epidermis to papilloma – there is always this transition. I think this paragraph can be removed.

Response 14: We revised the paragraph to compare our findings to previous reports as below. The discussions for two histopathological findings were removed. “Ramsauer et al. showed that an EcPV2-associated penile mass could be histopathologically distinguished from early stage, including hyperplasia and papilloma, to the late stage, including in situ carcinoma and SCC [19]. In our case, pathological findings of papilloma with proliferating epidermal cells, granular signals with ISH, and p53-positive deep epithermal cells were observed especially in the head of the mass compared to the stalk part, which was consistent with the previously reported findings of papilloma [19]. The nuclear staining images in ISH for E6/E7 have also been observed in human papillomavirus-associated tumors, which is thought to be associated with the life cycle of human papillomavirus; DNA episome synthesis increases and single-stranded human papillomavirus (HPV) occurs in the productive phase of HPV [20].”

Comment 15: Line 182. I completely agree the main transmission route of EcPV2 is sexual. As before, this virus has been shown to be more common in geldings and has been detected in horses that have never bred and in foals.

Response 15: We revised the sentence: “while sexual transmission is thought to be one of the transmission routes of EcPV2 [7,21].”

Comment 16: Line 185. This is very speculative. It seems there is only 1 other Japanese EcPV2? However, there are lots from elsewhere – until you have more Japanese EcPV2 genomes making any sort of firm conclusions about the origin of the virus cannot be supported.

Response 16: The sentence was revised: “Several genetic types of EcPV2 may already exist among horses in Japan, despite limited information on EcPV2 in the country.”

Comment 17: Line 197. The previous report was a single case – therefore the statement that mares had poor outcomes doesn’t make sense. Most of the conclusions of this paragraph seem to be based on a single papilloma in a stallion and a single SCC in a mare. This isn’t justified.

Response 17: We removed the sentence “However, our previous report showed that mares with EcPV2-associated SCC in Japan had poor outcomes [15]. The pathogenicity and malignancy of EcPV2-associated masses should be studied by adding case studies.”

Comment 18: Line 206. Every papilloma has variable severity of hyperplasia – these are not two different histological findings as stated.

Response 18: We removed the part of the sentence “in which two different histopathological findings were observed in a single mass.”

Reviewer 2 Report

Comments and Suggestions for Authors

This study reported a new case of Equus caballus papillomavirus 2 (EcPV2)-associated penile mass in an imported 22-year-old stallion with a history of breeding in Japan and abroad. Two histopathological regions, epidermal hyperplasia and papilloma, were observed in the mass. The virus was sequenced and found to be not closely related to a previously reported strain from a Japanese mare. The authors suggest that various types of EcPV2 might already exist among horses in Japan, based on the limited information available. While the overall study is simple and straightforward, additional analyses and rewriting could improve the readership and rigor of this interesting case report.

Here are some comments for this study:

Line 59-67: More detailed information on the “22-year-old retired Thoroughbred stallion” is needed to better guide future studies. For example, how long was he kept in each country/place? When was he imported to Japan? When did he start to breed with mares in Japan? When was this penile lesion observed? Is he still being used for breeding after the surgery?

Fig 2: In addition to E6/E7 RNA, L1 RNA would indicate whether infectious virions are present in this lesion, a potential indicator of virion shedding and transmission. Including a negative control (or explaining why one was not included here) would be helpful. Arrows in the images to denote the different cells discussed would also be beneficial. Please use a square (like in Fig 1) to indicate the region focused on in A and B.

Line 126: Did you separate the “viral DNA” from the DNA in the mass, as the kit will extract DNA from the whole tissue (viral DNA + cell DNA)?

Line 143: Interesting! Since the stallion was never present in Italy, Australia, Belgium, or China, is it possible that the mares he bred with had some connection to these places?

Line 147: There are at least three possibilities: 1) he did not acquire the virus from mares in Japan; 2) he did acquire this subtype from Japanese mares with asymptomatic infections; and 3) He had a latent infection with this virus that was reactivated due to immune suppression (e.g., aging or stress). Nevertheless, he might have transmitted this virus to the mares he bred with since he was shedding virions before surgery.

Line 156: “Two lesions” — I thought there was only one lesion/mass with two distinct histological regions.

Line 177: “The genetic variation of EcPV2 was not strongly related to geographic origin,” seems unjustified unless you have a large enough sample size to prove this.

Line 187-193: Do you have a complete record of this stallion? Those records would be extremely helpful to answer your postulation here. If this stallion has bred with other mares, you might want to collect some vaginal swabs to screen for viral DNA to determine whether he has transmitted the virus via sexual transmission to them; or check for serum conversion in them if you cannot find detectable viral signals, an indication of a previous infection that was cleared. Of course, he might have acquired the virus from the previous mares he bred with. Either way, this would help to trace back the source of this virus.

Line 196: “Histopathological findings also suggested that our patient’s tumor remained in an early stage compared with the SCC in a previous report.” Although the lesion is not malignant, subclinical infections can transmit infectious virions to others.

Line 209: This is quite common for any cancers; they are heterogeneous in nature but will be diagnosed with the highest histological category observed in any region of a certain tissue.

Comments on the Quality of English Language

The paper requires editing to more effectively convey its content.

Author Response

Thank you for kindly reviewing and supportive suggestions.

The revised manuscript highlighted the modified points according to the reviewers’ comments in yellow and highlighted the grammatically modified points in red font.

We revised the manuscript as follows.

  • Comment 1: Line 59-67: More detailed information on the “22-year-old retired Thoroughbred stallion” is needed to better guide future studies. For example, how long was he kept in each country/place? When was he imported to Japan? When did he start to breed with mares in Japan? When was this penile lesion observed? Is he still being used for breeding after the surgery?

Response 1: The sentences were revised: “He was used as a racehorse in Ireland until three-year old, then served as a stallion in the United States and Australia for 5 years, followed by 2 years in the United States and Argentina. He was then imported to Japan, where he served as a stallion for 10 years. The patient mated with dozens to three hundred mares per year until 3 years before the resection of the mass. After retiring as a stallion, he remained in Japan. Depigmented plaques had been observed in his penis for around 12 years before the resection of the mass, and finally a 4 cm × 4 cm mass on the top of the penis was observed without pain or deterioration of the general condition (Figure 1).”

  • Comment 2: Fig 2: In addition to E6/E7 RNA, L1 RNA would indicate whether infectious virions are present in this lesion, a potential indicator of virion shedding and transmission. Including a negative control (or explaining why one was not included here) would be helpful. Arrows in the images to denote the different cells discussed would also be beneficial. Please use a square (like in Fig 1) to indicate the region focused on in A and B.

Response 2: Thank you for your suggestion. We did not test ISH for L1 RNA for this patient. A negative control was difficult to include because we could not obtain the normal penile tissue. We will consider them for further study.

The squares were added in Figures 2A and 2B. The inset showing the granular signal of the cell was added to Figure 2C. Other photographs have not been revised because positive cells are currently visible.

  • Comment 3: Line 126: Did you separate the “viral DNA” from the DNA in the mass, as the kit will extract DNA from the whole tissue (viral DNA + cell DNA)?

Response 3: The kit used for DNA extraction did not separate viral DNA from cell DNA. We used the papillomavirus-specific primers for PCR and sequencing, so we understand that the contamination of cell DNA could be ignored.

The sentence was revised: “DNA was obtained using the QIAamp DNA Mini Kit (QIAGEN, Hilden, Germany) according to the manufacturer’s instructions.”

  • Comment 4: Line 143: Interesting! Since the stallion was never present in Italy, Australia, Belgium, or China, is it possible that the mares he bred with had some connection to these places?

Response 4: Our case was used as a stallion in Australia and had contact with Australian horses, however, our case did not have direct contact with horses in Italy, Belgium, or China. Followings could be suggested for this result.

  1. Mares contacted in his breeding in Japan or abroad were imported from Italy, Belgium, or China. Unfortunately, it would be difficult to verify the history of mated mares.
  2. The genetic information of EcPV2 has been reported from a limited number of countries and the genetic variation we could see did not accurately reflect the genetic variation of EcPV2 worldwide.

The first point was discussed in lines 168–169. For the second point, the sentences in the Discussion section were revised: “In this study, the phylogenetic tree obtained based on E6 and L1 could not elucidate the geographic origins of strains, due to the limitation that the genetic information of EcPV2 has been reported from a limited number of countries and may not accurately reflect the global genetic variation of EcPV2.”

  • Comment 5: Line 147: There are at least three possibilities: 1) he did not acquire the virus from mares in Japan; 2) he did acquire this subtype from Japanese mares with asymptomatic infections; and 3) He had a latent infection with this virus that was reactivated due to immune suppression (e.g., aging or stress). Nevertheless, he might have transmitted this virus to the mares he bred with since he was shedding virions before surgery.

Response 5: The sentences were added to the Discussion section: “The case might have experienced the EcPV2 infection prior to onset, and EcPV2 could have been reactivated due to immune suppression or aging.” and “The breeding with asymptomatic carriers in Japanese mares may be one of the transmission routes of our isolate.”

  • Comment 6: Line 156: “Two lesions” — I thought there was only one lesion/mass with two distinct histological regions.

Response 6: The results and discussions for two histopathological findings were revised in whole manuscripts.

The sentence was revised: “The histopathological findings contribute to the understanding the pathological status of EcPV2-associated penile neoplasms.”

  • Comment 7: Line 177: “The genetic variation of EcPV2 was not strongly related to geographic origin,” seems unjustified unless you have a large enough sample size to prove this.

Response 7: The sentence was revised: “In this study, the phylogenetic tree obtained based on E6 and L1 could not elucidate the geographic origins of strains, due to the limitation that the genetic information of EcPV2 has been reported from a limited number of countries and may not accurately reflect the global genetic variation of EcPV2.”

  • Comment 8: Line 187-193: Do you have a complete record of this stallion? Those records would be extremely helpful to answer your postulation here. If this stallion has bred with other mares, you might want to collect some vaginal swabs to screen for viral DNA to determine whether he has transmitted the virus via sexual transmission to them; or check for serum conversion in them if you cannot find detectable viral signals, an indication of a previous infection that was cleared. Of course, he might have acquired the virus from the previous mares he bred with. Either way, this would help to trace back the source of this virus.

Response 8: Thank you for your constructive advice. Unfortunately, it would be difficult to verify the history of mated mares, especially abroad. In Japan, the inspection of mares needs to be carefully considered to ensure that the racehorse breeding industry is maintained. However, your suggestion would However, we believe that your advice gives important information on the status of EcPV2 in Japan, which we will consider in future studies.

  • Comment 9: Line 196: “Histopathological findings also suggested that our patient’s tumor remained in an early stage compared with the SCC in a previous report.” Although the lesion is not malignant, subclinical infections can transmit infectious virions to others.

Response 9: The sentence was added: “Furthermore, EcPV2 in the genital region could be transmitted through breeding, posing a risk of EcPV2 infection for other horses..”

  • Comment 10: Line 209: This is quite common for any cancers; they are heterogeneous in nature but will be diagnosed with the highest histological category observed in any region of a certain tissue.

Reponse 10: The results and discussions for two histopathological findings were revised in whole sentences. We removed the part of the sentence “in which two different histopathological findings were observed in a single mass.”

Reviewer 3 Report

Comments and Suggestions for Authors

In the present case report, the Authors presented the histopathological and virological findings of the first reported case of an EcPV2-associated penile mass in Japan. Using the histopathological examination (RNA in situ hybridization targeting the E6/E7 region and immunohistochemical approach) and whole-genome sequencing of the mentioned penile neoplasm of the 22-year old stallion, two histopathological findings (epidermal hyperplasia and papilloma), were observed, depending on the location of the mass, and the EcPV2 virus was shown to be genetically close to foreign strains and different from the previously reported strain from a Japanese mare, respectively. The mentioned virological finding suggest that that various types of EcPV2 might already exist among horses in Japan and even though the malignancy of the tumour was low, the penile neoplasm would still be the an obstacle to breeding.

The used Methods were chosen well, as the Authors performed the histopathological and virological examination of the mass, the results are well-presented and concise and the discussion places them within the framework of previous research. However, I still have some minor comments for the Authors consideration. 

Minor comments

1) Please include the whole genome sequence of EcPV2 in the NCBI database, in addition to the DDBJ database.

2) Please add bootstrap values next to the branches in the phylogenetic tree of L1 nucleotide sequences (Supplementary Figure S1).

3) Since the whole genome sequence of EcPV2 was obtained, it could be interesting to the Readers if the Authors provided a more thorough description of the mentioned virus, e.g. describing the individual genes and their characteristics.

4) Please make sure that the manuscript is proofread by the English native speaker

Comments on the Quality of English Language

Minor English editing required.

Author Response

Thank you for kindly reviewing and supportive suggestions.

The revised manuscript highlighted the modified points according to the reviewers’ comments in yellow and highlighted the grammatically modified points in red font.

We revised the manuscript as follows.

  • Comment 1: Please include the whole genome sequence of EcPV2 in the NCBI database, in addition to the DDBJ database.

Response 1: After the sequence become available for public in DDBJ database (when the report is published), the sequence will become also available from NCBI database.

  • Comment 2: Please add bootstrap values next to the branches in the phylogenetic tree of L1 nucleotide sequences (Supplementary Figure S1).

Response 2: The figure of L1 sequences (Supplementary Figure S1) was revised adding bootstrap values.

  • Comment 3: Since the whole genome sequence of EcPV2 was obtained, it could be interesting to the Readers if the Authors provided a more thorough description of the mentioned virus, e.g. describing the individual genes and their characteristics.

Response 3: Our isolate has amino acid mutation in L1 and amino acid deletion in E6. Both were not characteristic of our isolate. The effects for L1 or E6 were not evaluated; in silico, mutations in these areas do not seem to affect function.

The sentence was added: “The sequence obtained from E6 of our isolate showed an amino acid deletion (p. Arg72del), which was not typical for our isolate, and its impact on the function of E6 was unknown.”

  • Comment 4: Please make sure that the manuscript is proofread by the English native speaker

Response 4: English editing was done by editage (www.editage.com). The revised manuscript was edited again. English proofreading certificate was attached.

Round 2

Reviewer 1 Report

Comments and Suggestions for Authors

Thanks for making the changes. Just note equine PVs are subdivided into 'types' not 'species' as stated. 

Comments on the Quality of English Language

A couple of minor changes to language needed.